# Iron Deficiency Anemia: Efficacy and Limitations of Nutritional and Comprehensive Mitigation Strategies

**DOI:** 10.3390/nu14142976

**Published:** 2022-07-20

**Authors:** Shashi Bhushan Kumar, Shanvanth R. Arnipalli, Priyanka Mehta, Silvia Carrau, Ouliana Ziouzenkova

**Affiliations:** Department of Human Sciences, The Ohio State University, Columbus, OH 43210, USA; drshashikumar81@gmail.com (S.B.K.); arnipalli.1@buckeyemail.osu.edu (S.R.A.); mehta.487@buckeyemail.osu.edu (P.M.); carrau.4@buckeyemail.osu.edu (S.C.)

**Keywords:** iron deficiency anemia, ferritin, C-reactive protein, RDA

## Abstract

Iron deficiency anemia (IDA) has reached epidemic proportions in developing countries and has become a major global public health problem, affecting mainly 0–5-year-old children and young women of childbearing age, especially during pregnancy. Iron deficiency can lead to life-threatening loss of red blood cells, muscle function, and energy production. Therefore, the pathogenic features associated with IDA are weakness and impaired growth, motor, and cognitive performance. IDA affects the well-being of the young generation and the economic advancement of developing countries, such as India. The imbalance between iron intake/absorption/storage and iron utilization/loss culminates into IDA. However, numerous strategic programs aimed to increase iron intake have shown that improvement of iron intake alone has not been sufficient to mitigate IDA. Emerging critical risk factors for IDA include a composition of cultural diets, infections, genetics, inflammatory conditions, metabolic diseases, dysbiosis, and socioeconomic parameters. In this review, we discuss numerous IDA mitigation programs in India and their limitations. The new multifactorial mechanism of IDA pathogenesis opens perspectives for the improvement of mitigation programs and relief of IDA in India and worldwide.

## 1. Anemia: Classifications and Global Significance

Anemia is a condition in which hemoglobin (Hb) concentration is characterized by reduced concentrations of hemoglobin (Hb) in the blood below cut-off levels, and/or a diminished number of red blood cells (RBC or reticulocytes) [1]. The cut-off values of Hb concentrations are specified by World Health Organization (WHO) for major populations; however, ethnicity, gender, and sex as well as the pathophysiological status may change these criteria [1]:–children 6 months to 4 years: Hb < 11.0 g/dL,–children 5 to 11 years: Hb < 11.5 g/dL,–children 12–14 years: Hb < 12.0 g/dL,–adult > 15 years old: men Hb < 13.5 g/dL, women Hb < 12 g/dL

Anemia is recognized as a global public health problem by WHO [2]. Particularly in developing countries like India the prevalence of anemia among adolescents comprises from 10 to 20%, according to a recent study [3].

The etiology of anemia is classified based on the number and morphology of RBC using the reticulocyte index (RI) and the mean corpuscular volume (MCV) [4]. The major anemia classes comprise hyperproliferative (RI ≥ 2.5) and hypoproliferative (RI < 2.5) anemia. Hypoproliferative anemias divide into microcytic (MCV < 80 femtoliters (fL)), normocytic (MCV 80–100 fL), and macrocytic (MCV > 100 fL) anemia types [5]. Anemia is not a disease, but a presentation of an underlying condition [6]. It is frequently classified based on biological mechanisms [7]. Dietary deficiencies play a causal role in the development of many types of anemia [8]. Numerous minerals, including iron, cobalt, magnesium, and micronutrients, including vitamin A, folate, B6, B12, and other B vitamins are directly required for the formation of RBC, Hb synthesis, and iron absorption, as well as antioxidant defense and cellular energetics [9,10]. The dietary etiology of anemia is multifactorial; nonetheless, iron plays a central role in oxygen delivery by Hb in RBC. The decrease in the total content of iron in the body leads to iron deficiency. Severe iron deficiency can reduce Hb levels, decrease erythropoiesis, and result in hypoproliferative microcytic anemia, which is defined as iron deficiency anemia (IDA) [5,11]. Iron deficiency [12,13] underlies the majority of anemia cases, affecting 1.6 billion people, which comprise 24.8% of the world population [8]. IDA is the major anemia condition around the globe that affects more than 6 million individuals of all segments of the population and contributes to approximately 840,000 deaths [14]. Particularly in India, more adolescents (10–20% of the studied adolescent population [3]) were diagnosed with IDA compared to other countries (2–6%) [15].

In this review, we will discuss IDA in the context of other nutritional deficiencies, environmental conditions, and other biological, socioeconomical, and political factors contributing to its prevalence. We focus on disparities in India presenting specific aspects responsible for IDA. Epidemiology, prevention, and mortality related to IDA as well as various strategies and policies adopted by the government of India to mitigate this major health problem will be discussed.

## 2. Disparities in India: Relevance of IDA Risk Factors

The required iron content is obtained from daily consumption of plant and animal food diets. In India, dietary iron intake is grossly inadequate and insufficient in most of the states, i.e., meeting less than 50% of recommended dietary allowances (RDA), according to the 2003 survey by the National Nutrition Monitoring Bureau (NNMB) [16]. However, this investigation also revealed the paradoxical relationship between iron intake and IDA. For example, 23 mg/d iron intake in women in Gujarat was associated with 55% anemia prevalence, whereas low 11 mg/d iron intake in women in Kerala was associated with only 33% anemia prevalence (Figure 1A) [16]. This difference was observed among all subsets of the population (Figure 1B). This lack of correlation between the prevalence of iron intake and IDA suggests that other risk factors substantially contribute to the development of IDA and could be relevant for the prevention of this pathology. Although the phenomenon of IDA risks in Gujarat has not been systematically investigated, other factors than iron intake, such as vitamin A deficiency, can play an important, or even causative role in iron availability in the body (Figure 1A) [10].

Vitamin A deficiency contributes to IDA through regulation of hematopoiesis, splenic iron concentrations, and mobilization of iron stores from tissues [18,19]. Hepcidin and ferroportin were identified as mechanistic targets that inversely correlated with vitamin A levels in circulations [20,21]. The net effect of increase in hepcidin/ferroportin under vitamin A deficient conditions is: (1) release of iron from liver stores into circulation; (2) stimulation of iron absorption; (3) concomitant accumulation of iron in macrophages, and (4) suppression of erythropoiesis, leading to anemia [22]. Inflammatory conditions of infections and metabolic diseases trigger similar changes in vitamin A-dependent regulation of iron metabolism [19]. Although this pathophysiological link between vitamin A deficiency and IDA is documented, the biomarkers of vitamin A-dependent IDA have not been established for use in clinical settings.

Recently, vitamin D has been proposed as an alternative pathway regulating hepcidin/ferroportin axes; however, vitamin D deficient status appears to have less pronounced effect than vitamin A deficiency [21]. The emerging link between vitamin D and IDA requires further investigation due to the role of iron in the enzymatic conversion of vitamin D into its hormonally-active metabolites [23].

Iron availability has two interdependent components: (1) accessibility of nutrients and minerals meeting recommended dietary allowance (RDA) levels and (2) effective absorption of these nutrients. Risk factors act cumulatively to increase the prevalence of IDA in the population (Figure 2). The risk factors affecting these processes are discussed below.

### Risk Factors for IDA in Different Populations

Although RDA for iron has been established for different age and sex sub-populations, dietary intake to achieve RDAs in different populations varies, because it is influenced by the environment in geographical locations, socioeconomic factors, individual pathophysiology, as well as cultural traits of diet composition and consumption (Figure 2). The major population characteristics altering RDA requirements are age, sex-related functions, and inherited disorders such as hemoglobinopathies [24]. Growth and development increase the demand for iron and other nutrients, particularly in pregnant and lactating women, and children. These groups represent high-risk populations, which rapidly develop IDA in response to inadequate dietary intake of iron. The health consequences of IDA in children and adolescents are stunted growth, impaired cognitive development, and poor mental and motor performance [25]. ID even without anemia causes fatigue and detrimental effects on cognition and mental health in women of childbearing age [26].

In adults, IDA impairs memory capacity, attention span, and cognitive development, resulting in up to 30% of impairment in physical work performance [27,28]. Maternal death is three times higher in women with IDA than without this condition [29]. In developing countries, the rapid growth of the population without adequate socioeconomic protection leads to disparities in nutrition. Consequently, more than half of women 15–49-year-old (53%) and almost one-quarter of 15–49-year-old men (22.7%) are anemic (Table 1). The problem is augmented in the high-risk populations. The prevalence of IDA is on the rise amongst children 6–59 months of age (69.5%), married women (55.3%), breastfeeding mothers (63.2%), and pregnant women (52%), according to the periodic surveys by the National Family Health Survey (NFHS II III and IV) [30,31] and the National Nutrition Monitoring Bureau [32,33]. In India, six out of every ten 6–59-month-old children are anemic [30,31]. IDA in the young population results in marked economic losses up to 8% of the Indian GDP [27,28].

Young women stand apart as the most disadvantaged population due to the bleeding-related physiological iron losses during menstruation and labor, in addition to repeated pregnancies and lactation that increase iron demand [35]. The availability of medical care is a critical socioeconomic factor in preventing excessive blood loss during menarche, pregnancy, and labor by adequate timing of the umbilical cord clamping [36,37]. Inadequate medical care is ultimately responsible for 12.8% of maternal deaths due to IDA in Asia [38]. Socioeconomic status is a major determinant affecting diet quality and iron availability across the life span.

Different states of India have a similar prevalence of IDA, according to the district levels household survey (DLHS). However, the reduced socioeconomic status in rural vs. urban areas has significant impact on the prevalence of IDA in India [31]. The prevalence of IDA is higher in rural compared to urban areas based on NFHS IV and NFHS V surveys [31]. Diet quality, the proportion of rural vs. urban population, and iron RDA depend on physical environments in the geographical regions of India [12,13]. High altitude, temperature, and humidity affect oxygen binding to Hb; therefore, iron RDA requirements need to be increased in these regions [12,13]. However, staple foods of India cannot adequately address iron intake, particularly among vegetarians.

In developing countries, low dietary intake is related not exclusively to the low content of iron in the diet, but also due to its poor iron bioavailability. Meta-analysis revealed that inadequate concurrent intake of dietary iron from Indian staple foods, especially rice and millets, is one of the major etiological factors for the high prevalence of IDA in India [12,13,17]. Dietary factors contributing to iron malabsorption are listed in Table 2. In risk subpopulations, ID is augmented by poor iron reserves at birth and frequent infections in childhood [36,37]. Absorption of iron is compromised by chronic inflammation, inflammatory bowel disease (IBD), congestive heart failure, chronic kidney disease, obesity, cancer, rheumatoid arthritis [39,40,41,42,43], dysbiosis, as well as parasitic infections including malaria [16]. Subsets of the population affected by genetic diseases could be more sensitive to reduction in iron intake than the general population. For example, subjects with sickle cell anemia disorder, characterized by a mutation of the β-globin chain of Hb (HbS) could rapidly develop IDA despite apparently adequate dietary intakes [44,45]. Hyacinth et al. [45] proposed to establish specific RDA requirements for subjects with sickle cell anemia, particularly during the pregnancy and growth phase. Similar adjustments for RDA are probably needed to account for pathophysiological and environmental factors in specific populations.

The conditions associated with IDA are shown in Table 2. Figure 3 demonstrates multiple mechanisms underlying the development of iron deficiency and augmenting its progression to IDA. Recent reviews discussed the interplay among gastrointestinal factors [68], iron and regulation of oxidative stress and immunity [69], as well as vitamin D metabolism [23], among other signaling events. Several cultural and biological traits could worsen iron availability, intake, and absorption in the Indian population, including lack of bioavailable heme iron in the diets of vegetarians, representing 33% of adults in South Asia vs. 2.4% of the US population [70]. Moreover, the defective iron absorption is amplified by the gastric pH difference in the Indian population (~pH 3.4) vs. Western country populations (pH 2.5) [71]. Severe problems with iron bioavailability related to a deficiency in micronutrients and severity of infectious and inflammatory diseases have been recently reviewed by Nair and Iyengar [16] and reported by others [72]. Chronic iron deficiency also has maternal determinants [73]. The amount of iron stored mainly depends on the gestational length period and birth weight of the baby. However, in India approximately 7.4 million infants are born either premature or with less than 2.5 kg birth weight [73]. Despite this established role, the contribution of absorption, dysbiosis, and infection are rarely analyzed in the context of IDA prevalence in India. The incidence of many infectious and parasitic diseases continues to increase in India [74]. These factors represent strategical targets for the prevention and treatment of IDA.

## 3. Multifactorial Etiology of IDA Poses Challenges for Diagnosis and RDA

Prevention and treatment of IDA depend on the accurate and sensitive diagnosis of iron deficiency and its transition to IDA. However, inflammation, diets, environmental stressors, and metabolism influence the iron levels in the blood and the development of reliable biomarkers of iron status in blood remain challenging. In deficient states, the serum levels of Hb and iron bound to ferritin are reduced [39,75,76]. The total iron-binding capacity (TIBC) in serum is also increased due to impaired iron binding to the transport protein (apo-transferrin) and increased soluble transferrin receptor (sTfR) resulting in a low transferrin saturation [39,75,76]. Although ferritin is used as an indirect indicator of total iron concentration in the body, its concentrations are increased during inflammatory conditions, in response to infections, or malignant tumors [75,76,77,78]. Thus, ferritin alone remains an unreliable diagnostic marker during inflammation. The emerging trend in diagnostics is using multiple markers to measure iron status. The ratio of serum ferritin to C-reactive protein (CRP) levels was proposed as an indicator of iron status during inflammation [79]. Alternatively, sTfR is used as an indicator of the stored intracellular iron levels, which is independent of inflammation [12]. The low serum iron and ferritin levels with increased TIBC are also used in diagnosing ID [11]. Recently, it has been demonstrated that the ratio of sTfR/log ferritin, also known as (sTfR-F index), is considered a reliable marker for biochemical identification of iron deficiency [12]. Diagnosis by blood iron levels and ferritin levels [80] is incomplete for patients with sickle cell anemia, which is diagnosed in newborns in India by analyzing blood cell count [81]. Raman spectroscopy [82], HbS by electrophoresis, high-performance liquid chromatography (HPLC), and isoelectric focusing [83] have been proposed as an emerging diagnostic approach to distinguish between IDA and sickle cell anemia. A great potential for differential diagnosis of IDA in individuals with β-thalassemia has been demonstrated by the Trait Neural network model compared to traditional analyses, such as a complete blood cell count or serum ferritin concentrations [84].

In India, in epidemiological and clinical settings, ID is diagnosed based on conventional biochemical markers such as blood Hb concentrations as well as serum levels of iron, transferrin, transferrin saturation, ferritin, and/or sTfR (Table 1). Currently, IDA is defined as a reduced concentration of blood Hb or hematocrit (33%) along with iron stores depletion, indicated by ferritin levels in blood [85]; however, blood loss can provoke an increase in immature RBC, reticulocytes, produced by bone marrow [86]. IDA manifested morphologically in blood as hypochromic and microcytic mature RBC and decreased levels of reticulocytes [85]. The advanced IDA diagnostics utilizes an automated analyzer of mean corpuscular volume (MCV), percentage of mature hypochromic erythrocytes (HyPom), red cell distribution width (RDW), and reticulocyte changes in Hb concentration (CHr) and size (Ret Y). These indices are more sensitive than the Hb level alone. Measurements of iron status were the basis for RDA recommendations; however, the availability of new diagnostic tools of iron status based on multiple biomarkers may require RDA re-evaluation for populations at risk in India and the development of RDA recommendations for subpopulations in different regions of India. 

## 4. Recommended RDA of Iron for Indians

The requirements of RDA in different populations were recently revised by the Indian Council of Medical Research (ICMR 2010) [87]. This RDA for iron is based on a factorial approach, accounting for the basal loss of iron, and additional requirements for pregnancy, lactation, and growth without considering the poor bioavailability of iron from habitual Indian diets. The new revised iron RDAs in 2010 are elucidated in Table 3 [87]. Although the revised RDAs aimed to address iron status in India, the specifics of diet and the presence of multiple factors limiting absorption were not accounted for. Thus, it is not validated if the current RDA adequately addresses IDA and prevents the development of IDA in different regions of India. Moreover, in the absence of socioeconomic measures, these RDA goals for iron intake cannot be achieved in the high-risk populations of IDA.

## 5. Efficacy of IDA Prevention Programs in India

The WHO has recognized anemia in developing countries as a worldwide problem. The World Health Assembly Resolution 65.6 in 2012 has approved an action plan for infant, maternal, and child nutrition to curtail anemia prevalence in women of reproductive age by half by 2025 to ensure healthy lives for all by 2030 [88]. Following this WHO program, the prevalence of anemia, including IDA, among women of reproductive age is on the decline in countries such as Nepal, from 65% to 34% in 8 years; in China, from 50.0% to 19.9% in 19 years; in Bhutan from 54.8 to 31.1% in 12 years; in Sri Lanka, from 59.8% to 31.9% in 13 years; and in Vietnam, from 40.0% to 24.3% in 14 years [89]. Thus, with substantial government commitment, a reduction in the prevalence of IDA is considered feasible between 2012 and 2025.

India was the first country to recognize the problem of IDA and the Union Ministry of Health launched the National Nutritional Anemia Prophylaxis Program (NNAPP) in 1970 during the 5-year plan in all the states of India [90]. It was aimed at preventing anemia in the most vulnerable populations, i.e., pregnant and lactating mothers as well as preschool children between one and five years of age. Under the National Nutrition Policy program (1993), expectant, nursing mothers, as well as acceptors of family planning [91] were given one tablet of 100 mg elementary iron and 500 μg of folic acid per day. Infants between the ages of 6–60 months were given iron supplements in the liquid formulation of 20 mg elemental iron and 100 μg folic acid per day for 100 days. Children (6–10 years old) received 30 mg of elemental iron and 250 μg of folic acid per day for 100 days. However, adolescents of 11–18 years received supplements at the same dosage as adults [90,92]. The NNAPP is currently being taken up by the Maternal and Child Health (MCH) Division of the Ministry of Health and Family Welfare, which is a part of the Reproductive Child Health (RCH) program. Despite this and other programs and continuous efforts at national and international levels, various community-based studies from 1950–2002 have revealed an increase in the prevalence of IDA in India [93]. These increasing trends of IDA could be the result of administrative malpractices including many policies, programmatic gaps, inadequate medical coverage, or ignorant tendencies in Indian society due to cultural customs and beliefs [93]. Furthermore, planners and implementers are discouraged from designing interventions addressing multifactorial solutions beyond health. Unfortunately, these trends were also persistent in the post-independence period, which is known as the ’Green Revolution’ in 1965. The government of India launched the Green Revolution with the help of a renowned geneticist Dr. M.S. Swaminathan. This revolution changed the country’s food-deficient economy into one of the world’s leading agricultural nations for the efficient production of rice and wheat using pesticides, herbicides, and fertilizers [94]. Following the Green Revolution, rice and wheat became the main cereal crops and changed dietary patterns. Traditionally, these cereals with low iron content are consumed in large quantities and have become a primary, but insufficient, source of iron for the Indian population. The consumption of these main cereal crops led to a dramatic reduction in the consumption of coarse cereals, such as jowar (*Sorghum bicolor*), bajra (*Pennisetum typhoideum*), maize (*Zea mays*), and ragi (*Eleusine coracana*), from 23% of cereal calories in 1983 to 6% among rural and urban populations (data for 2011) [95]. Original Indian millets, such as bajra and ragi, have iron levels of 14–17 mg/100 g, but they are only consumed in some parts of the country [36]. The decline in the cultivation and consumption of traditional coarse cereals with high iron content led to a decline in dietary iron content in the Indian diet, which coincides with an increased incidence of IDA in women and children [96]. In recent interventions, increased consumption of coarse cereals, particularly millets, reduced the prevalence of IDA in Indian women and children [97]. The recent meta-analysis of 22 studies has shown that regular consumption of millets effectively increased Hb levels by 13.2% and serum ferritin by 54.7% [97]. NITI Aayog of the Govt. of India took into account these findings in December, 2021 and developed a Statement of Intent with the United Nations World Food Program (WFP) acknowledging the exchange on mainstreaming the use of millets to improve the dietary availability of iron. The year 2023 was announced as the International Year of Millets.

## 6. Current Prevention and Control Strategies against IDA in India

### 6.1. Programs

Numerous governmental programs focusing mostly on nutritional and population risks were developed and implemented to combat IDA from 1963 to 2021.

A National Nutrition Policy was implemented by the government of India in the Department of Women and Child Development, Ministry of Human Resource Development, Government of India to address the problem of undernutrition/malnutrition [98]. The implementation of this multi-sectoral strategy for women and child development involves numerous steps:Setting up an intersectoral coordination mechanism at the center, state, and district levels;Advocacy and sensitization of policymakers and program managers;Intensifying activities to address micronutrient malnutrition;Providing nutrition information to people;Establishing a monitoring system of nutrition and mapping at the community, district, and state levels;Establishing district-wide disaggregated data on nutrition.

This policy has two strategies, namely direct strategies (short-term goals) and indirect strategies (long-term goals). The policy envisioned undertaking direct interventions by fortification of essential foods with iron, popularization of low-cost nutritious foods and strengthening the NNAPP with the introduction of iron supplementation for adolescent girls.

This policy expands the safety net to at-risk populations, such as children, adolescent girls, and women. Indirect strategies include ensuring food security, improvement in dietary patterns through production and demonstration, improvement in purchasing power, performing land reforms, and providing essential health and nutrition education [99]. Programs were coordinated with the health and nutrition surveillance [99].

Integrated Child Development Services (ICDS) is one of the world’s largest and most unique programs for early childhood development, which was launched on 2 October 1975 in 33 (4 rural, 18 urban, and 11 tribal) blocks [100]. This program has four different components:Early Childhood Care Education & Development (ECCED)Health ServicesCommunity Mobilization Awareness, Advocacy and Information, Education, and CommunicationCare and Nutrition Counselling

Under this program, supplementary feeding support was provided for 300 days per year to children six years of age or younger as well as pregnant and nursing mothers [100]. This program intended to bridge the caloric gap and control nutritional anemia in disadvantaged communities. However, the positive impact on children’s nutritional status was not observed because this program’s implementation was not uniform across different areas of India and did not account for other IDA risk factors. Thus, complementary nutrition (CN), growth monitoring and growth faltering, convergence and coordination, community participation, and administrative corrections represent urgent corrective strategies to strengthen the ICDS program in different areas.

The National Program for Nutritional Support to Primary Education (NP-NSPE) scheme was launched on 15 August 1995 as a centrally sponsored scheme by the government of India. It was further revised in 2004 and 2006. In October 2007, NP-NSPE was renamed as ‘National Program of Mid-Day Meal’, which is popularly known as the ‘Mid-Day Meal Scheme’ with the inclusion of milk in mid-day meals for children [101]. However, in September 2021, the Mid-Day Meal Scheme was again renamed ‘PM POSHAN’ or Pradhan Mantri Poshan Shakti Nirman. The objective of the scheme is to invigorate the effectiveness of primary education by impacting the nutritional status of children in primary class schools countrywide. In newly revised guidelines, the nutritional value of a hot, fresh-cooked midday meal has been augmented from 300 to 450 kcal. The protein content has increased from 8 g to 12 g. This scheme has also included adequate quantities of micronutrients such as iron, folic acid, and vitamin A in meals [101]. This scheme has many additional potential benefits, such as engaging children from disadvantaged schools, improving the regularity of meal intake, nutritional values, and quality of meals, as well as socialization. However, current statistical analysis revealed that 42.5% of the children under 5 years of age remain underweight; moreover, India has an alarmingly low Global Hunger Index, ranked 101 among 161 countries in 2021 [102]. This rank has decreased compared to the 94 rank in 2020 [102]. Therefore, despite numerous revisions, the Mid-Day Meal remains ineffective and requires substantial improvement.

A multi-pronged 12 × 12 initiative has been launched to address IDA in adolescents across the country, with the target to achieve a Hb level of 12 mg/L by the age of 12 years by 2012 [103]. This initiative has been supported by numerous professional organizations such as the Government of India, the World Health Organization, the Indian Council of Medical Research, UNICEF, and the Federation of Obstetric and Gynecological Society of India [104,105]. The main objective was to invigorate health and nutrition education, as well as provide weekly supplementation with iron and folic acid. Unlike other programs, this initiative includes periodic deworming practices along with appropriate immunization to ensure parasite control.

The Rajiv Gandhi Scheme for Empowerment of Adolescent Girls (RGSEAG) was initiated by the Ministry of Women and Child Development in April 2011. This is one of the centrally sponsored programs from the Government of India known as the SABLA Scheme [106]. SABLA aimed at improving nutritional and health status in adolescent 11–18-year-old girls, along with other goals, such as upgradation of home, life, and vocational skills [106]. Initially, it was implemented in 200 districts of all states/union territories, which had a lower composite index than other districts across the country. This program replaced the previous Nutrition Program for Adolescent Girls (NPAG) and Kishori Shakti Yojana (KSY). Under this program, free food grains are provided to underweight adolescent girls (kg per beneficiary per month). Under SABLA, each adolescent girl must be given at least 600 kcal and 18–20 g of protein and adequate micronutrients at RDA levels for 300 days per year [106]. SABLA has been implemented in 553 districts out of 728 total districts in India since 2019, according to the web portal SAG-RRS [107]. However, despite its commendable efforts to improve nutrition in adolescent girls throughout India, SABLA has fallen short of achieving its desired outcomes. The weaknesses of SABLA include poor program monitoring, non-utilization of funds, gaps between intended and achieved targets, limited capacity of Anganwadi workers (AWWs), and lastly, imposing restrictions on parents.

Anemia Chale Jao-Nischay-2007 was initiated by the Federation of Obstetric and Gynecological Society of India (FOGSI) in the light of the “illiteracy of health” in women to improve the detection and management of IDA and other forms of anemia. In fact, 80% of females remain unaware of the basic health parameters, i.e., height, weight, Hb levels, and blood group. The objective of FOGSI was to eliminate IDA by educating Indian women in every household to know their weight, height, blood group, and Hb levels by the end of 2007 [108]. After the diagnosis of IDA, women were provided with iron tablets for 1 month [105]. However, the lack of follow-up within this program limits its success.

Saloni Swasthya Kishori Yojna is a part of a USAID-funded pilot project implemented between 2004–2010. This in-school project encompasses nutrition and reproductive health interventions for 7- to 14-year-old girls in Uttar Pradesh, Uttarakhand, and Jharkhand. Under this initiative, school drop-out adolescent girls need to be identified with the promotion of the use of iron and folic acid syrup (IFA, Table 4) for supplementation in adolescent groups. However, this program had similarly poor monitoring as other programs, therefore, the magnitude of IDA in girls was not reduced.

The National Iron Plus Initiative (NIPI) aimed to address the major nutritional aspects of IDA in India revealed by the National Family Health Survey-3 (NFHS-III) [109,110]. It focuses on the high-risk groups for IDA (Table 1), i.e., nearly 58% of pregnant women, 50% of non-pregnant and non-lactating women, 56% of adolescent girls (15–19 years), 30% of adolescent boys and around 80% of among children under 3 years of age and 70% below 5 years of age (Table 1). This program defines a minimum service of packages for the treatment and management of IDA. IFA supplementation was implemented concurrently for pregnant and lactating women, and children in the age group of 6–60 months along with the inclusion of new-age groups such as school children (5–10 years) and women of the reproductive age (Table 4) [92]. This NIPI program is expected to be executed by more than 700,000 community health volunteers termed Accredited Social Health Activists (ASHA) [111]. At present, there are many constraints to the effective implementation of this program. Some of the constraints are:(i)Assessment of asymptomatic pathologies affecting digestion of supplements: the hidden and silent diseases beyond diarrhea are not elucidated.(ii)Poor compliance with IFA supplementation often resulted from insufficient counseling regarding the benefits and possible minor side effects of IFA.(iii)Accessibility of IFA supplements to providers: IFA supplementation is commonly hindered by the transportation of IFA from district warehouses to the providers, such as community health centers.(iv)Procurement and monitoring by ASHA: Workers have been given the major responsibility to implement all the activities of the NIPI program. The Ministry of Health and Family (MoHFW) and other ministries allocate ASHA responsibilities for the execution of NIPI with newer other programs without additional personnel. The procurement and distribution of IFA are disrupted by these inadequate logistics.(v)Impaired coordination of programs by various ministries, such as the Ministries of Health and Family Welfare, Women and Child Human Resource, Tribal Affairs, Rural Development, and Urban Development: Therefore, the Government of India has emphasized and taken imperative steps to improve interdepartmental coordination, supervision, and monitoring milestones for the success of this program [110].

Prime Minister’s Overarching Scheme for Holistic Nutrition is known as the POSHAN Abhiyaan was launched in March 2018 to reduce IDA. Abhiyaan, is also known as the National Nutrition Mission (NNM), which was launched on 8 March 2018 on International Women’s Day in Jhunjhunu at Rajasthan. The flagship of this program is to improve nutritional outcomes for children, pregnant women, and lactating mothers. This program aims to achieve the following targets by 2022:Reducing stunting by 2% annually;Reducing under-nutrition by 2% annually;Reducing anemia by 3% annually;Reducing low birth weight by 2% annually.

Anemia Mukt Bharat 6 × 6 × 6 Strategy was designed at the NITI Aayog, which is the prime public policy think tank of the Government of India. The Anemia Mukt Bharat strategy has some overlapping goals with the PHOSAN Abhiyaan and National Nutrition Strategy. It aims to reduce the prevalence of IDA by 3% per year between 2018 and 2022 among children, adolescents and the WRA group, i.e., 15–49 years. The 6 × 6 × 6 strategy to combat IDA uses six beneficiaries, six interventions, and six institutional mechanisms (Table 5) [112].

### 6.2. Food Fortification

Fortification of staple foods, such as rice, milk, and salt, is a common practice to augment the content of essential vitamins and minerals to improve their nutritional content and provide a public health benefit with minimal health risk. In 2004, Micronutrient Initiative (MI) was launched in India and supported the installation of a double fortified salt (DFS) manufacturing facility at the Tamil Nadu Salt Corporation (TNSC) plant. Micronutrient Initiative (MI, 2012) in India has developed two products, Anuka and Vita-Shakti. The 0.5 g Anuka food supplement includes iron (3 mg), and vitamins A (300 IU) and C (30 mg). The Vita-Shakti supplement has iron (7 mg), vitamin A (500 μg), and folic acid (50 μg). These two products aim to improve the intake of vitamins and minerals of young children 6–24 months of age (MI Toolkit Products for Emergencies and Food Security Programs 2012). Apart from these two products, MI has also developed lozenges fortified with vitamin A, iron, and other nutrients, often called Nutri-candies or Nutri-lozenges, to protect children who have no access to centrally processed and accessible foods and fortified foods [105,113]. The Global Alliance for Improved Nutrition (GAIN) organization supports the Naandi Foundation to supply fortified rice to schools for children in the state of Andhra Pradesh. The provision of fortified foods is a part of the Government of India’s Midday Meal Scheme to augment enrollment, retention and attendance while improving the nutritional status of students in primary classes at the Naandi Foundation [114]. The major concern about this fortification is the combination of prooxidant iron with vitamin A, which is susceptible to oxidation by reduced iron [115]. This combination of iron with vitamin A influences the stability of the product [116] and its nutritional value.

In less affluent areas in India, micronutrient deficiencies are associated with a staple diet based on starch-enriched foods with limited access to foods from animal sources. Dietary diversification, which includes increasing the consumption of meats, dairy products, fruits, and vegetables at the household level, has been proposed as the major intervention strategy for anemia [117,118,119]. These interventions would increase the variety and quantity of micronutrient-rich food to combat micronutrient deficiencies. Recent studies suggest that these programs help in reducing the prevalence of IDA [117,118,119].

### 6.3. Current Efficiency of IDA Mitigation Programs

The implementation of the nutrition-focused programs utilized India’s self-sufficiency in food production [120], which is a key component of programs tackling undernutrition and malnutrition [121]. In India, based on the latest fifth round of survey NFHS V in 2019–2021, IDA alarmingly increased in all risk groups compared to NFHS IV [31] in 2015–2016 (Figure 4) [17]. It affects 67.1% of children below the age of 3 years, 52% of expectant mothers, and 57% of women aged between 15 to 49 years in India [17]. Thus, IDA among women and children has become worse in most states/union territories during the last half-decade (NFHS). This result was unexpected because the statistical analysis has demonstrated improved health care outcomes for other diseases in the last four years in India [17] except for a steady rise in IDA in women and children despite many mitigation programs funded by the state (discussed above).

The reasons for the failure of numerous programs to eradicate IDA could depend on global and Indian-specific issues. Analysis of data from NHFS IV in India identified undernutrition, highlighted by BMI below normal levels, as a factor increasing risks for COVID-19 [122]. Although the role of IDA in susceptibility to COVID-19 was not explored in this study, Gujarat was among the states with the highest incidence of both IDA and COVID-19 [122]. Indeed, IDA patients are more susceptible to COVID-19 compared to a control group (61% vs. 45%, respectively), according to a recent observational study in 71 patients [123]. The SARS-CoV-2 virus appears to mimic the action of hepcidin by increasing the levels of ferritin in tissues and circulation, which results in ID and low Hb levels in serum [124]. In addition, SARS-CoV-2 and other viruses utilize iron for replication [125]. Moreover, iron sequestered during inflammation [126] exacerbates IDA. Although a medication decreasing the intracellular content of iron shows promise in the treatment of viral infections [127], it also could lead to IDA. Iron supplementation should be used with utmost precaution during viral infection; however, it is a critical measure to address IDA in the post-infection period. Therefore, COVID-19 could be a global issue contributing to the increased incidence of IDA in India despite mitigation programs.

The India-specific issues increasing IDA prevalence percolates in socioeconomically lagging communities especially in scheduled castes, tribes, and among children born to uneducated women and their low compliance. However, the critical issue is to develop a new program addressing the multifactorial causes of IDA and ensure their implementation for all risk groups for IDA.

### 6.4. Strategical Directions to Improve Interventions to Combat IDA

The following intervention or strategies need to be addressed in any program aimed at improving general well-being, and especially the improvement of iron status. Multiple factors to include in these strategies are elucidated below [105,128,129,130]:Poverty reduction;Ensuring food security;Improvement in antenatal services;Creating awareness program among mothers;Improvement of accessibility to diversified diets;Promotion of better care and feeding practices i.e., exclusive breastfeeding;Food fortification and food-based approaches;Prevention and treatment of infectious diseases such as malaria and tuberculosis, etc.;Promoting safe water, sanitation, and hygiene (WASH);Parasitic disease control programs, especially for manifestation of helminths;Integration with other micro-nutrient control programs;Monitoring and evaluation of programs;Implementing innovative programs;Intersectoral coordination;Advocacy and social communication;Strengthening the surveillance system;Continuous monitoring of new research and its incorporation into the program.

The most effective strategies include mass food fortification with iron, targeted iron supplementation program/intervention, control of hookworm and malaria, education to the general public about iron-rich sources of plant- and animal-based foods. They continue to be used to combat IDA among Indians. However, there is an urgent need for operational and system research to increase compliance among beneficiaries. One of the main operational obstacles is passive administration and a lack of communication and development among public-private partnerships. Apart from this, further research is needed to optimize the dose and frequency of supplementation (daily vs. weekly), and absorption/bioavailability in clinical and community set-ups. More work is needed to identify a cost-effective solution for diets and supplements to improve their accessibility to populations at risk for IDA and public compliance with their use. These measures need to be multi-factorial and tailored to Indian regional cultural traditions as well as the dietary, genetic, environmental, and pathophysiological factors contributing to IDA. The socioeconomic measures will play a key role in the implementation of these tasks. Therefore, experts from various fields such as clinical research, biochemistry, microbiology, genetic and computational engineering, imaging, and modeling should work jointly with socioeconomic specialists and political organizations to develop a new innovative diagnostics and monitoring programs and controlled robust implementation to combat IDA in hundreds of millions of patients worldwide.

## Figures and Tables

**Figure 1 nutrients-14-02976-f001:**
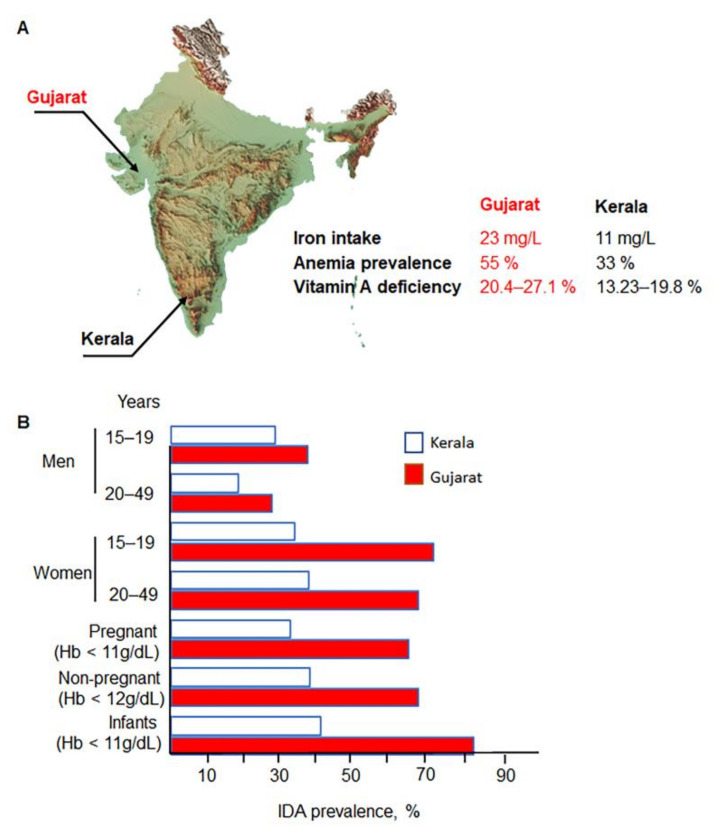
Prevalence of IDA is higher in Gujarat compared to Kerala despite higher iron intake in Gujarat state. (**A**) Gujarat has a higher prevalence of vitamin A deficiency than Kerala. Kerala presents a lower iron intake than Gujarat. (**B**) Prevalence of IDA in Gujarat in all population subsets compared to Kerala. Data are from 2019–2021 [17].

**Figure 2 nutrients-14-02976-f002:**
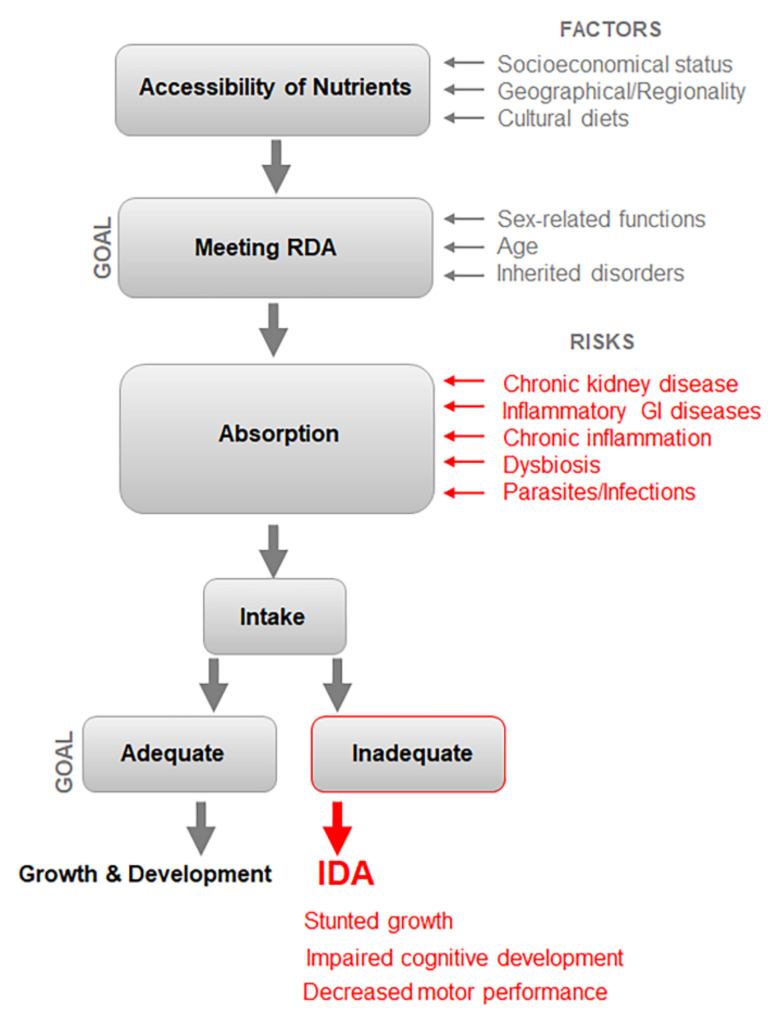
The pathogenesis mechanism of IDA depends on both accessibility of nutrients and their absorption. Socioeconomic and cultural factors influence the dietary amount and its composition. Diet and other environmental factors impact both accessibility and absorption, which increase RDA and pose risks of inadequate intake. Gastrointestinal (GI) inflammation and dysbiosis caused by metabolic and genetic diseases, parasites, and infections lead to inadequate intake of iron and other critical nutrients and, consequently, IDA. The mild and moderate characteristic features associated with IDA are fatigue, weakness, koilonychia, pica, and pallor, ultimately resulting in the major impairments of growth, motor, and cognitive performance.

**Figure 3 nutrients-14-02976-f003:**
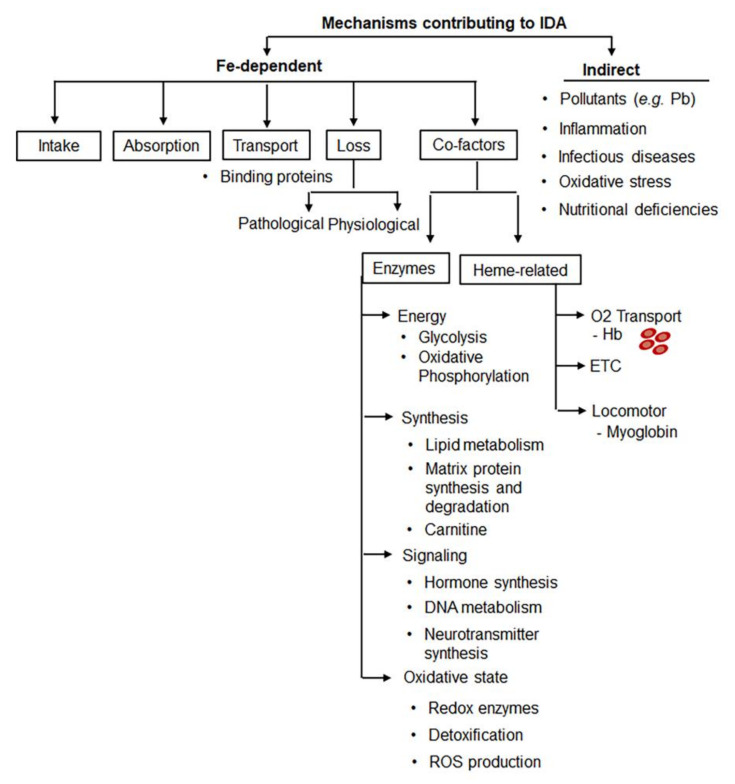
Crosstalk between mechanisms participating in IDA. Pathogenesis of IDA involves iron (Fe)-dependent mechanisms and other factors regulating iron metabolism and erythropoiesis (e.g., essential nutrients, minerals and their metabolites). Iron deficiency, a contributing factor of IDA, can be caused by defective intake, absorption, and transport of iron and/or be a result of iron loss. Digestive, inflammatory, and blood-loss associated pathologies (Table 2) as well as sickle cell anemia and other genetic diseases involve these pathways. Iron deficiency compromises the function of iron-dependent co-factors leading to defective energy production, nervous system, and hormonal regulation dysregulating Hb synthesis and erythropoiesis leading to IDA. Inflammation, infectious diseases, and oxidative stress increase demand for iron and energy production as well as impair absorption of iron and other nutrients, thereby promoting IDA. ETC, electron transfer chain.

**Figure 4 nutrients-14-02976-f004:**
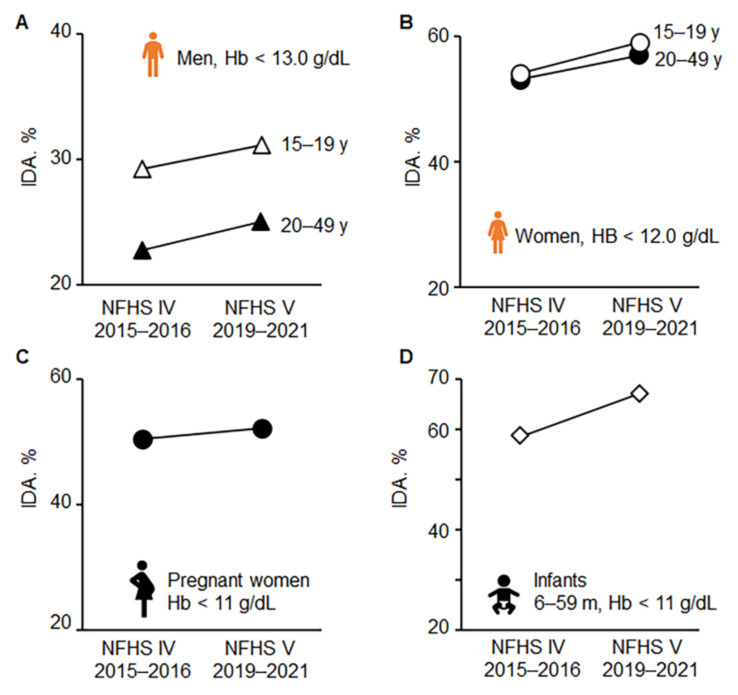
Increased prevalence of IDA from 2015 to 2021 in adolescent and adult men (**A**) and women (**B**) as well as in pregnant women (**C**) and infants (**D**) in India based on NFHS IV and V surveys. NFHS, National Family Health Survey.

**Table 1 nutrients-14-02976-t001:** Prevalence of IDA in India. NFHS means National Family Health Survey (NFHS). Source: Nutrition Atlas, [34].

Group	Age (Years old)	Hb Level (g/dL)	NFHS II (%)	NFHS III (%)	NFHS IV (%)
Children	6–35	<11	74.2	78.9	58.4
Childbearing women	15–49	<12	51.8	56.2	53.0
Pregnant women	15–49	<11	49.7	57.9	50.3
Men	15–49	<13	--	24.3	22.7

**Table 2 nutrients-14-02976-t002:** Common pathophysiological conditions associated with IDA. The interplay between mechanisms participating in the development of IDA is described in Figure 3. UTI: urinary tract infection; NSAID: non-steroidal anti-inflammatory drug; H2: histamine 2.

Blood Loss-Related Conditions	Malabsorption	Anemia of Chronic Diseases	Genetic Disorders
Digestive tract Colorectal cancer [46] Gastric carcer [47] IBD [37] Peptic ulcers [48] Angiodysplasia [49] Parasites [49] *Entamoeba histolytica* *Giardia intestinalis* *Ascaris lumbricoides* *Plasmodium falciparum* Gynecological causes [50] Menstruation Labor Delayed umbilical cord clamping Surgeries Urinary tract (hematuria): UTI Bladder cancer Renal cancer Hemodialysis Respiratory tract: Epistaxis Hemoptysis Drugs: Aspirin and other NSAID [11] Vitamin E toxicity	Celiac disease [51] Gastrectomy *Helicobacter pylori* [52] Bowel resection Atrophic gastritis Bypass gastric surgery Bacterial overgrowth Interaction with food elements [53]: Tea, coffee, Calcium Flavonoids, Oxalates, Phytates Millet Wheat Pica syndrome [54] Pagophagia Drugs [55,56]: PPI H2 blockers	Congestive heart failure [49] Cancer [57] Chronic kidney disease [58] Rheumatoid arthritis [28] Obesity [59,60] IBD [61,62]	Iron-refractory iron deficiency anemia [58] Divalent metal transporter deficiency anemia [63] Fanconi anemia [64] Pyruvate kinase deficiency [65] Thalassemia [66] Sickle cell anemia [44,67]

**Table 3 nutrients-14-02976-t003:** RDA for iron in India. Y-year old; m- month. Source: RDA for Indians, Indian Council of Medical Research, 2010, India [64].

Group		Age Range	Body Weight (kg)	Requirement (mg/kg/d)	Absorption (Assumed %)	RDA (mg/d)
Man		15-lifelong	60	14	5	17
Woman	Overall	15-lifelong	55	30	8	21
	Pregnant	15–49 y	55	51	8	35
	Lactating	15–49 y	55	23	8	25
	Infants	0–6 m	5.4	46	--	--
		6–12 m	8.4	87	15	15
Children		1–3 y	12.9	35	5	9
		4–6 y	18.0	35	5	13
		7–9 y	25.1	31	5	16
Adolescents						
	Boy	10–12 y	34.3	31	5	27
		13–15 y	47.6	34	5	32
		16–17 y	55.4	25	5	28
	Girl	10–12 y	35.0	38	5	27
		13–15 y	46.6	29	5	27
		16–17 y	52.1	25	5	26

**Table 4 nutrients-14-02976-t004:** NIPI intervention and regime. IFA-syrup supplement of iron and folic acid (1 mL containing 20 mg iron/100 mg folic acid). The Mother and Child Protection card (MCP) is an entitlement card and counselling and family empowerment tool, which would ensure tracking of the mother and child cohort for health, nutrition, and development purpose; the MCP card has the potential to create awareness, facilitate community dialogue and generate demand for uptake of vital services being provided as the first contact point between a pregnant woman and the health system. The MCP card is maintained by ASHA and Auxiliary Nurse Midwifery (ANM).

Age Group	Intervention/Dose	Regime	Service Delivery
6–60 m old	IFA syrup	IFA biweekly from 6 to 60 m old Deworming, children >12 m old	ASHA/ANM: inclusion in MCP card
5–10 y old	Tablet (45 mg iron/400 mg folic acid)	Weekly from 5 to 10 y old Deworming, biannually	Teacher for aged-school children AWC: children out of the school
10–19 y old	Tablet (100 mg iron/500 µg folic acid	Weekly from 10 to 19 y old Deworming, biannually	Teacher: school children AWC: children out of the school
Pregnant/lactating women	Tablet (100 mg iron/ 500 mg of folic acid)	Daily from 14 to 16 weeks of gestation. Repeated for 100 days during post-partum	ANC/ANM/ASHA: inclusion in MCP card
Women in reproductive age (WRA)	Tablet (100 mg iron/500 mg of folic acid)	Weekly throughout the reproductive period	FHW during a home visit for contraceptive distribution

**Table 5 nutrients-14-02976-t005:** Anemia reduction Target for 2022 under Anemia Mukt Bharat 6 × 6 × 6 strategy. WRA, the White Ribbon for Safe Motherhood, India.

	Percentage (%) Prevalence of Anemia	
Group	NFHS IV	Target *
Children (6–59 months-old)	58	40
Adolescent girls (15–19 years-old)	54	36
Adolescent boys (15–19 years)	29	11
WRA	53	35
Pregnant women	50	32

* Reduction of IDA prevalence by 3% per year from baseline data.

## Data Availability

Not applicable.

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
