# Peer review of "Iron Deficiency Anemia: Efficacy and Limitations of Nutritional and Comprehensive Mitigation Strategies"

_nutrients, 2022, doi:10.3390/nu14142976_

Round 1

Reviewer 1 Report

This review article on iron deficiency anaemia by Shasi Bhutan Kumar and colleagues is quite extensive and comprehensive.

The following minor points require attention:

page 2/18 when mentioning Vitamin A deficiency and IDA, the authors need to add a sentence to explaining the patho-physiological mechanism underlying this link.

Furthermore, since in India  haemoglobinopathies including Sickle Cell Anaemia are also common, an additional paragraph about the differential diagnosis with the same should be added. For this scope, you can use these references:

Sickle cell disease and dental treatment A Piccin, P Fleming, E Eakins, E McGovern, OP Smith, C McMahon.  J Ir Dent Assoc 54 (2), 75-79 

Insight into the complex pathophysiology of sickle cell anaemia and possible treatment A Piccin, C Murphy, E Eakins, MB Rondinelli, M Daves, C Vecchiato, ... European journal of haematology 102 (4), 319-330

Author Response

Reviewer 1 Comments and Suggestions for Authors

This review article on iron deficiency anaemia by Shasi Bhutan Kumar and colleagues is quite extensive and comprehensive.
            We would like to thank this reviewer for his/her fruitful suggestions.

The following minor points require attention:

page 2/18 when mentioning Vitamin A deficiency and IDA, the authors need to add a sentence to explaining the patho-physiological mechanism underlying this link.

We added this information Lines 84-92: ‘Vitamin A deficiency contributes to IDA through regulation of hematopoiesis, splenic iron concentrations, and mobilization of iron stores from tissues [15, 16]. Hepcidin/ferroportin were identified as mechanistic targets that inversely correlated with vitamin A levels in circulations [17, 18]. The net effect of increase in hepcidin/ferroportin under vitamin A deficient conditions is: 1) release of iron from liver stores into circulation; 2) stimulation of iron absorption; 3) concomitant accumulation of iron in macrophages, and 4) suppression of erythropoiesis, leading to anemia [19]. Inflammatory conditions of infections and metabolic diseases trigger similar changes in vitamin A-dependent regulation of iron metabolism [16]. Although this pathophysiological link between vitamin A deficiency and IDA is documented, the biomarkers of vitamin A-dependent IDA have not established for use in clinical settings’.

Furthermore, since in India haemoglobinopathies including Sickle Cell Anaemia are also common, an additional paragraph about the differential diagnosis with the same should be added. For this scope, you can use these references:

Sickle cell disease and dental treatment A Piccin, P Fleming, E Eakins, E McGovern, OP Smith, C McMahon.  J Ir Dent Assoc 54 (2), 75-79 

Insight into the complex pathophysiology of sickle cell anaemia and possible treatment A Piccin, C Murphy, E Eakins, MB Rondinelli, M Daves, C Vecchiato, ... European journal of haematology 102 (4), 319-330

We added this Sickle Cell Anaemia with these references to the list of conditions associated with IDA in Table 2. We also added the description in:

Lines 172-179: ’Subset of population affected by genetic diseases could be more sensitive to reduction in iron intake than general population. For example, subjects with sickle cell anemia disorder, characterized by mutation of the β-globin chain of Hb (HbS) could rapidly develops IDA despite apparently adequate dietary intakes [41, 42]. Hyacinth et al [42] proposed to establish specific RDA requirements for subjects with sickle cell anemia, particularly during the pregnancy and growth phase. Similar adjustments for RDA are probably needed to account for pathophysiological and environmental factors in specific populations’.

Lines 227-234: ‘The diagnosis by blood iron levels and ferritin levels [82] is incomplete for patients with sickle cell anemia, which is diagnosed in newborn in India by analyzing blood cell count [83]. Raman spectroscopy [84], HbS by electrophoresis, high-performance liquid chromatography (HPLC), isoelectric focusing [85] has been proposed as an emerging diagnostic approach to distinguish between IDA and sickle cell anemia. A great potential for differential diagnosis of IDA in individuals with β-Thalassemia has been demonstrated by Trait Neural network model compared to traditional analyses, such as complete blood cell count or serum ferritin [86].'

Reviewer 2 Report

Dear Editor, many thanks for the opportunity to revise this manuscript. In the paper, the authors gave a picture of pathophysiology, epidemiology and risk factors of iron deficiency anemia as well as proposed interventions to decrease burden of this condition in Indian population. The topic is interesting and up-to-date, but I have many major concerns to be addressed:

-Introduction and title: clarify if the focus of the paper is IDA in general or IDA due to nutritional deficiencies only.

-Global significance of nutrition-related anemia: the title needs probably some adjustment and the flow of information is sometimes confusing and not clear. You should first define anemia in general and then collocate iron deficiency anemia (that actually lacks a definition in this paper). Iron deficiency can be the only cause of anemia (pure IDA) or a contributing factor of anemia in the mixed forms. Please refer to WHO classification of anemia in general (putting the cut-offs at the beginning of the paragraphs and specify which age groups cut offs are related to) and then describe the main types of anemia and definition of IDA. The same concern regards the difference of anemia prevalence between India and the rest of the world. 

-Line 3:  "anemia is not a diagnosis" could be changed into "anemia is not a disease", since referenced WHO criteria are used to diagnose anemia.

- Line 36-37: clarify which means "80-90%" (it can't be the overall prevalence of anemia in the world).

Line 40-41: the interpretation of percentages 27 and 6 % can be misleading. You are referring to prevalence of anemia in adolescents or to prevalence of adolescents out of the global population?

-Line55 : change "has shown" with "was associated with".

-Line 67: write out the acronym RDA.

-Table 2: adjust formatting. Among diseases listed in Table 2, briefly discuss which ones have a major epidemiological impact in India ( in absence of study of associations with IDA, refer to overall prevalence of this conditions) and by which pathophysiological pathways they affect iron stores.

-Which is the role of Vitamin D deficiency (emerging factor for IDA) in India?

-Add a figure about the main mechanisms leading to IDA and diseases associated with that (among those listed in Table 2).

-Briefly describe the main treatments to improve iron stores in Indian population (apart from prevention or treatment of nutrition deficiencies). Which diseases are particularly bothersome in these populations? 

Author Response

Reviewer 2 Comments and Suggestions for Authors

Dear Editor, many thanks for the opportunity to revise this manuscript. In the paper, the authors gave a picture of pathophysiology, epidemiology and risk factors of iron deficiency anemia as well as proposed interventions to decrease burden of this condition in Indian population. The topic is interesting and up-to-date, but I have many major concerns to be addressed:

-Introduction and title: clarify if the focus of the paper is IDA in general or IDA due to nutritional deficiencies only.

We changed title to:’Iron Deficiency Anemia: efficacy and limitations of nutritional and comprehensive mitigation strategies’.

            We also clarified the focus of this paper

-Global significance of nutrition-related anemia: the title needs probably some adjustment and the flow of information is sometimes confusing and not clear. You should first define anemia in general and then collocate iron deficiency anemia (that actually lacks a definition in this paper). Iron deficiency can be the only cause of anemia (pure IDA) or a contributing factor of anemia in the mixed forms. Please refer to WHO classification of anemia in general (putting the cut-offs at the beginning of the paragraphs and specify which age groups cut offs are related to) and then describe the main types of anemia and definition of IDA. The same concern regards the difference of anemia prevalence between India and the rest of the world. 

We re-wrote the first paragraph to include the definition of anemia and clarify the differences between iron deficiency, IDA, and other type of anemias.

-Line 3:  "anemia is not a diagnosis" could be changed into "anemia is not a disease", since referenced WHO criteria are used to diagnose anemia.

            Thank you, this statement was changed.

- Line 36-37: clarify which means "80-90%" (it can't be the overall prevalence of anemia in the world).

We changed this number based on the recent study in Sci Rep 2022 May 17;12(1):8197. doi: 10.1038/s41598-022-12258-6

Line 40-41: the interpretation of percentages 27 and 6 % can be misleading. You are referring to prevalence of anemia in adolescents or to prevalence of adolescents out of the global population?

            We clarified that population represent adolescents.

-Line55 : change "has shown" with "was associated with".

We changed this verb.

-Line 67: write out the acronym RDA.

The acronym was explained.

-Table 2: adjust formatting. Among diseases listed in Table 2, briefly discuss which ones have a major epidemiological impact in India ( in absence of study of associations with IDA, refer to overall prevalence of this conditions) and by which pathophysiological pathways they affect iron stores.

Formatting is adjusted. We briefly discussed in text the specific factors contributing to IDA in India.

-Which is the role of Vitamin D deficiency (emerging factor for IDA) in India?

We briefly discussed this link in the revised manuscript, Line 94-98: ’ Recently, vitamin D has been proposed as an alternative pathway regulating hepcidin/ferroportin axes; however, vitamin D deficient status appear to have less pronounced effect than vitamin A deficiency [21]. The emerging link between vitamin D and IDA required further investigation due to the role of iron in the enzymatic conversion of vitamin D into its hormonally-active metabolites [23].’

-Add a figure about the main mechanisms leading to IDA and diseases associated with that (among those listed in Table 2).

The new Figure 3 was added.

-Briefly describe the main treatments to improve iron stores in Indian population (apart from prevention or treatment of nutrition deficiencies). Which diseases are particularly bothersome in these populations? 

 We included this description: Lines 193-207.

Round 2

Reviewer 2 Report

The manuscript significantly improved and it is now worth of publication.